# Age Assessment in Children and Adolescents by Measuring the Open Apices in Teeth: A New Sardinian Formula

**DOI:** 10.3390/dj10040050

**Published:** 2022-03-22

**Authors:** Enrico Spinas, Giorgia Melis, Nicoletta Zerman, Stefano De Luca, Roberto Cameriere

**Affiliations:** 1Department of Surgical Sciences, Sport and Dental Traumatology Research Center, University of Cagliari, 09124 Cagliari, Italy; 2Department of Surgical Sciences, Dental Prosthetic Division, University of Cagliari, 09124 Cagliari, Italy; giorgiamel51@gmail.com; 3Department of Surgical, Dentistry, Maternal and Child Sciences, University of Verona, 37129 Verona, Italy; nicoletta.zerman@univr.it; 4Panacea Cooperative Research S. Coop., 24402 Ponferrada, Spain; stfndlc@gmail.com; 5AgEstimation Project, Department of Medicine and Health Sciences “Vincenzo Tiberio”, University of Molise, 86100 Campobasso, Italy; roberto.cameriere@unimol.it

**Keywords:** forensic sciences, age estimation, dental age, open apices, children, Sardinia

## Abstract

Age estimation in children is fundamental in both clinical and forensic fields. The aim of this study was to evaluate the accuracy of the Cameriere’s European and Italian formulae for age estimation in Sardinian children and adolescents, a genetically isolated population. A sample of 202 orthopantomograms of healthy Sardinian children and adolescents (100 females and 102 males) aged between 6 and 17 years was retrospectively evaluated. The seven left mandibular teeth were assessed with the Cameriere’s European and Italian formulae. The teeth with closed apex (N0) were counted and, in the teeth with open apex, the distance between the inner sides was calculated. All variables showed a significant and negative correlation with age except N0 and g. Sex (g), the variables s, N0, and the first-order interaction between them, contributed substantially to the age measurement (*p* < 0.001). Although the value of x5 had a low prediction level, it generated the following multiple linear regression formula, specific for the Sardinian sample: Age = 10.372 + 0.469 g + 0.810 N0 − 1.079 s − 0.398 s ∙ N0 − 0.326 × 5. Only the Sardinian and European formulae allowed to obtain an acceptable interclass agreement (both the lower and upper >0.7). The results showed that the European formula could be accurate for assessing age in this sample of children and adolescents.

## 1. Introduction

Age estimation in children and adolescents is a topic of fundamental importance in paediatric and orthodontic/orthopaedic cases allowing dental experts to assess the most appropriate treatment of different dental malocclusions based on craniofacial growth in children [1].

In the last few years, the application of accurate age estimation methodologies on living subjects has gained an increasing importance in the forensic field [1,2,3]. In fact, age assessment can be relevant in order to establish the status of a minor individual, criminal liability, asylum request or adoption. In criminal proceedings, the completion of 14 years and 16 years of age is also of high importance. According to the Convention on the Rights of the Child [4], children below the minimum age of criminal responsibility cannot be considered responsible in criminal proceedings when they commit an offence. On the other hand, children that are at or above the minimum age but younger than 18 years are subjected to all those justice procedures required for a child. 

In accordance with the international and interdisciplinary Study Group on Forensic Age Diagnostics (AGFAD), in case of legal justification for X-ray examinations without a medical indication, estimating age through the analysis of hand-wrist and dental radiographs is a simple and non-invasive procedure easily applied in the forensic daily practice. Furthermore, it is always in compliance with the principles of the best interest of the child (BIC) and the benefit of the doubt [4,5,6], the last one being a key concept in order to assess a specific age with the correct quantification of uncertainty [5].

According to Italian law, in order to achieve the highest accuracy of age estimation in forensics, a dental analysis by orthopantomography needs to be applied on a case-by-case basis [6].

The precision and accuracy of the results obtained by applying a dental approach depend on the age of the analysed subject, the accuracy being higher in children (<15 years) than in subadults (15–23 years) and adults (>23 years). In growing children, different biological indicators correlate better with the age of an individual [1]. However, the development of permanent teeth is more reliable than other indicators of age, such as the hand-wrist bones maturation, as the dental maturation and development is controlled more by genes than by environmental factors [1,7,8,9,10]. The dental age assessment methods are based on the identification of the stages of mineralisation in radiographic images, followed by their comparison with the standard stage to estimate the approximate age range (qualitative methods). In the last decades, the vast availability of intraoral or panoramic radiographs, in most orthodontic or paediatric practices, allowed the researchers to develop many qualitative methods/scores and charts for age estimation in numerous samples of children and adolescents from several countries. For instance, Demirjian et al. [11] developed one of the most used maturity score-based systems for the developmental stage of the seven left mandibular permanent teeth. However, its validation of various samples revealed a consistent overestimation of the real age [11,12]. 

In 2006, Cameriere et al. [13] developed a quantitative method for assessing age in children by measuring the open apices of tooth roots and the seven left permanent mandibular teeth were valued and measured.

Because most dental age assessing methods developed for children should be validated on samples of different origin, it is important to determine whether a country-specific model needs to be developed based on a larger reference sample. In 2007 [2], the same authors analysed a large sample of individuals from different European countries to generate a common European formula for age estimation in children. However, in these studies, the authors considered only data from a continental Italian population and other European populations, but they did not include any data from the Sardinian population [2].

The Sardinian population has always been the subject of interest by geneticists. Due to the presence of distinctive genetic frequencies, and other peculiar features, it can be considered as an outlier when compared to other Mediterranean populations and in studies of population genetics studies. Both insularity and some selective pressures, such as malaria, offered a unique opportunity to study the impact of natural selection on some current genetic characteristics of this population [14,15].

In the last 15 years, many studies have been carried out aimed at sequencing the whole genome which have tried to provide new data on the genetics of this population [15,16], all confirming the specificity of this group which appears to be certainly the richest in characteristics. Ancestral remains unchanged and are still found in the Sardinian population whose provenance of at least three out of four progenitors can be certified. Regardless of the time of initial population, the Sardinian population has certainly had pluri-millennial isolation times that have allowed for the development of a series of genetic peculiarities related to adaptation in an island environment [17,18], such as the unique incidence rates for autoimmune and haematological diseases, such as multiple sclerosis [19,20,21], type 1 diabetes [22], and also phenotypic morphological characteristics like stature [23].

The aim of this research was to test the validity of the European and Italian formulae [2,13] in a specific Sardinian population in order to determine if their genetic uniqueness and homogeneity may point out remarkable differences in its accuracy.

## 2. Materials and Methods

A well-balanced sample of 202 digital orthopantomograms (OPTs) of living Sardinian (Italy) subjects (102 males and 100 females), aged between 6 and 17, was evaluated retrospectively (Table 1). 

OPTs were obtained from the database of the School of Odontology (University of Cagliari, Sardinia, Italy), and collected during the period 2004–2020. The obtained images were taken from patients attending an orthodontic treatment or for diagnostic and therapeutic purposes. Only subjects with parents and grandparents born and residing in Sardinia, and with an official (local authority) declaration documenting their Sardinian residence since three generations prior, were included in the analysis. OPTs with lost or extracted single-rooted teeth, with fillings, with crown restorations and severe caries or other abnormal dental anatomy, previous orthodontic treatments, systemic diseases affecting the teeth development and potentially causing measurement difficulties, were removed from the analysis.

Since all the investigated subjects were under the age of 18, the consent to use data for research and educational purposes was obtained from the parents, and the Ethical approval was obtained from the University of Cagliari Human Ethics Committee. Besides, this study was carried out according to the ethical standards developed by the World Medical Association (WMA) for medical research with humans [24].

Patients’ identification number, sex, age, Sardinian residence, date of birth and date of collection were documented, but no further information relating to the patient’s identification was collected. Finally, to assure anonymity, each radiograph was assigned an alphanumeric serial code. 

### 2.1. Measurements

The methodology and the analyses performed here are fully explained in Cameriere et al. [13]. 

The number of teeth with completed root development and the apical ends of the roots totally closed (N0) was determined. All these teeth with open apices were also analysed. When teeth with one root were considered, the distance Ai, i = 1,…, 5, between the inner sides of the open apex was assessed. When teeth with two roots were examined, Ai, i = 6, 7, the sum of the distances between the inner sides of the two open apices was determined. In addition, the measurements Ai were normalised by dividing them by tooth length Li: xi = Ai/Li, i = 1,…, 7 in order to order to avoid any effect of magnification and angulation inherent in the X-ray’s image. Finally, dental age was assessed using the normalised measurements of the open apices of the seven left permanent mandibular teeth, xi, i = 1,…, 7, their sum (s), and the number (N0) of these teeth with root apex complete [13].

The selected OPTs were saved in JPEG format. In order to adjust brightness, contrast and grey scale, the image quality was amended using the specific tools in Adobe^®^ Photoshop^®^ CS4. A single observer measured the entire sample. 

In order to test the intra- and inter-observer agreement, two independent observers, both of them experienced with radiological imaging, evaluated 30 OPTs randomly selected one month after the first reading. 

### 2.2. Statistical Analysis

Each OPT was identified by a numerical ID to avoid observer bias, and the operators, consequently, did not know the age and sex of the subjects. The chronological age of each subject was calculated by subtracting the patient’s date of birth from the date of radiographic recording.

Intra- and inter-observer reproducibility of the measurements were evaluated using the intra-class correlation coefficient (ICC) based on the absolute agreement.

Following the procedure illustrated in Cameriere et al. [13], to obtain the age estimation as a function of sex and morphological parameters, a multiple linear regression model with first-order interactions was developed.

Pearson’s correlation coefficient (R) and the coefficient of determination (adjusted R^2^) were used to compare the results obtained by applying the new Sardinian formula and the already published formulae in the same Sardinian sample. Unlike R^2^, that never decreases no matter the number of variables we added to our regression model, adjusted R^2^ was selected because it takes into account the number of independent variables used for predicting the target variable. The adjusted R^2^ was used to increase only if the new variable improved the model more than would be expected by chance [25].

In addition, in order to test the accuracy of the new formula and the formula developed for the European populations, 41 Sardinian children and adolescents (19 males and 22 females), not previously included in any analysis, were studied. The chronological age as well as the estimated ones were evaluated, with an intra-class correlation based on an absolute agreement. All the statistical evaluations were carried out using the IBM SPSS 22.0 software program (IBM SPSS Statistics, Armonk, NY, USA). The significant threshold was set at 5%.

## 3. Results

No significant intra- and inter-observer differences between the re-examined sets of measurements were detected. The between-reader agreement was between 0.81 and 0.98, and within the reader agreement it was between 0.87 and 0.97. The correlation between age and teeth morphological variables was tested with a Pearson’s correlation analysis: all the variables showed a significant (*p* < 0.001) and negative correlation with age, except N0 (number of teeth with closed apices) and g (sex), which showed a positive correlation (Table 2).

The results showed that sex (g, males = 1, females = 0), the variables s (the sum of normalised measurements) and N0, and the first-order interaction between them contributed substantially to the fit (*p* < 0.001). To maintain a consistency with the previous approaches [2,9], the value of x5 (the ratio between apex width and height of the second premolar) was considered despite a low level of prediction (Table 2). All these variables were included in the regression model, yielding the following multiple linear regression formula (R = 0.952; adjusted R^2^ = 0.905):Age = 10.372 + 0.469 g + 0.810 N0 − 1.079 s − 0.398 s∙N0 − 0.326 × 5

The expected ages, based on the models developed on three populations, called “European” [2], “Italian” [13], and “Sardinian”, as reported in Table 3, were compared with the known chronological age of 41 Sardinian children not previously used for the regression analysis. The observed versus predicted comparison showed an optimal agreement for all the intra-class values (both for single and average measures, ICC > 0.8) and an excellent agreement when the Sardinian formula was used (ICC > 0.9). However, analysing the 95% CI of the obtained ICC values, only the Sardinian and the European formula allowed us to obtain an acceptable agreement (both lower and upper limits > 0.7), whereas the use of the Italian model does not provide satisfactory results (Table 3). 

The residual plots (Figure 1A–C) showed no obvious patterns in all the models. 

Hence, the statistical analyses and the diagnostic plots support the chosen models, with a better estimate obtained by applying the model developed in this work.

## 4. Discussion

The aim of this study was to test the validity of the Italian and European formulae developed by Cameriere et al. [2,13] in a Sardinian sample, considering its genetic peculiarity and given the evidence of a strong effect of genetic and environmental factors in the process of development and maturation of the teeth [26]. In fact, different molecular and genetic studies [26], performed on both human and animal specimens, have identified several genes that can regulate a specific stage of dental development, thus determining the differentiation process of each tooth.

Despite that, the number of studies investigating dental development in the Sardinian population is limited, with some contradictory results presented. In 1993, Diaz [27] conducted a study on a sample of 382 Sardinian children using the method of Moorrees et al. [28] in which emerged that, when compared with North American children, dental development in the Sardinian population was slowed down, especially in girls in all teeth except in the development of the third molar. On the contrary, Spinas et al. [29], after testing the applicability of the third molar index (I_3M_) to discriminate adults from subadults in a Sardinian sample, showed an earlier third molar development in males than in females, that is in accordance with the literature [30,31].

The results here, presented, showed that the ages derived applying the European formula are in optimal agreement with the chronological ages of the subjects, indicating a non-statistically significant effect of the genetic background on the applied methods. When considering the forensic and clinical literature, this is the first research about the usefulness of the Cameriere’s method [2,13] for assessing age in a Sardinian sample of children and adolescents. As already showed in a previous study [29], these methods could be helpful to estimate dental age and to elucidate quantitative traits of biomedical relevance, even in children of such a genetic isolate population.

Of course, the ages estimated with the formula generated from the Sardinian population indicated a better agreement compared to the one obtained applying the European formula, but the differences in the average value are not significant. In forensics, since the maximum and minimum values are considered, the range of the dental age estimates is deemed to be acceptable (see Table 3).

In addition, this new formula was tested using a set of data that are distinct (unseen data) from those used to train the first-stage model (training set), as showed by the comparison with the known chronological age of 41 Sardinian children not previously used for the regression analysis. This fact is fundamental to avoid overfitting and is essential for providing a forensic expert with precise and accurate estimation methods. If a different set of data was not used, a correct expression of “error of formula” could not be proposed. Since all the data are collected from patients attending an orthodontic treatment or for diagnostic and therapeutic purposes, what is defined as the “error of the model” is a true value, which highlights a normal bias or variation in the same formula. Thus, instead of talking about “error”, it would be better to refer, in this case, to a “quantification of uncertainty”.

The Italian model produces a lower agreement that, anyway, can be considered in an adequate range of uncertainty for assessing age in the clinical daily practice if the minimum and maximum values are considered. However, from a forensic point of view, and taking into account the maximum and minimum values (see Table 3), this quantification of uncertainty (margins of error) could not be in accordance with the ethical principles of the best interest of the child, autonomy, beneficence, non-maleficence, and justice, and the benefit of the doubt [4].

It is worth mentioning here; the results obtained in this study, such as in some previous studies [30,32], highlighted that the developmental stage of the second premolar (x5) has no significant correlation with age estimation. In fact, adding this parameter to the model did not increase the adjusted R^2^ value. Despite having a significant correlation in previous works, the x5 development has very little contribution to the final age estimation (weeks or months) when considering the children and adolescent’s age estimation.

In addition, this work indicates that N0 alone explains 87.1% of the variability in the model and is close to the same parameter found in the European formula. In fact, this parameter provided a significant contribution to the generated model having, in the Sardinian and European formula, a value close to 1, while in the Italian, it was 0.674 (see Table 4). In the last decade, several studies have been carried out in several countries in order to demonstrate the accuracy of the European formula; and the obtained results were diverse. In some populations [32,33,34,35], it was highly accurate to estimate ages in children, whereas for other populations (Indian [31], South African [36] and Chinese [37]), it needed to be integrated and a new population-specific formula had to be determined. However, it is evident that in the majority of these studies, the used sample was inappropriate to properly describe the different degrees of opening in the apices of teeth: when research includes a non-uniform number of subjects in the different age groups analysed, the bias (under- and overestimation) may arise and generate a phenomenon known as “age mimicry”.

In conclusion, for the first time, it has been demonstrated that the European formula can be used in the Sardinian population with a high level of precision and accuracy in the children’s and adolescents’ age estimates. It is important to highlight that these good results could be the consequence of a specific situation that is common to several studies: all the examiners were well informed about and trained in radiology, in general, and in this quantitative technique, specifically. When a new age estimation technique is being developed, training and calibrating observers with different experience is an essential part of the methodology in order to give the research high quality and precision. In the present study, high values of ICC were achieved for both repeatability and reproducibility of the technique developed by Cameriere et al. [13]. However, this fact emphasises the importance of using qualified personnel to assess age, and the priority of an appropriate level of training and experience.

One of the possible limitations of this study is the small sample size of children and adolescents. This fact should be highlighted as an important factor that caused the nonsignificant difference in developmental stages of each left lower tooth among European, Italian and Sardinian subjects. According to the literature [38], it has been demonstrated that differences in dental development should be considered in populations with heterogeneous origin when using national stage-based methods. In addition, genetic studies [39] confirm that the majority of the variations exist within a population made of different ethnic groups rather than between a large population. Thus, as a future perspective, a larger sample composed of a multi-ethnic balanced European cohort, including Sardinian children and adolescents, should be analysed in comparison with an Italian sample in order to verify if any genetic ancestral content can influence dental development.

## Figures and Tables

**Figure 1 dentistry-10-00050-f001:**
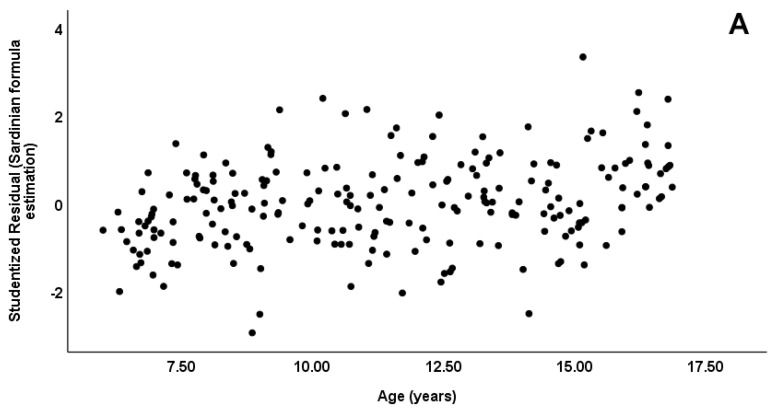
The residual plots ((**A**): Sardinian, (**B**): Italian and (**C**): European formula).

**Table 1 dentistry-10-00050-t001:** Distribution of the sample according to sex and age.

Age	Males	Females	Total
6–7	20	17	37
8–9	18	19	37
10–11	18	18	36
12–13	19	18	37
14–15	17	18	35
16	10	10	20
Total	102	100	202

**Table 2 dentistry-10-00050-t002:** Regression analysis predicting chronological age from the chosen predictors.

	Value	Standard Error	*t* Value	*p*
Constant	10.372	0.320	32.409	<0.001
g	0.469	0.143	3.271	0.001
N_0_	0.810	0.057	14.184	<0.001
s	−1.079	0.259	−4.156	<0.001
*s*∙N_0_	−0.398	0.102	−3.911	<0.001
x5	−0.326	1.028	−0.317	0.752

**Table 3 dentistry-10-00050-t003:** ICC values according to the comparison between the real and estimated age, using the Sardinian, Italian and European formula, respectively (* = acceptable agree; ** = optimal agree; *** excellent agree).

Formula		Intra-Class Correlation	95% CI Lower Limit	95% CI Upper Limit	*p*
Sardinian	Single measure	0.914 ***	0.760 *	0.962 **	<0.0001
Sardinian	Average measures	0.955 ***	0.863 **	0.981 ***	<0.0001
Italian	Single measure	0.852 **	0.130	0.955 ***	<0.0001
Italian	Average measures	0.920 ***	0.231	0.977 ***	<0.0001
European	Single measure	0.890 **	0.744 *	0.948 ***	<0.0001
European	Average measures	0.942 ***	0.853 **	0.973 ***	<0.0001

**Table 4 dentistry-10-00050-t004:** Comparison between the contributions of N0 values in the children’s age based on different regression models.

	Value	Standard Error	*t* Value	*p*
Sardinian	0.810	0.057	14.184	<0.001
Italian	0.674	0.040	17.02	<0.001
European	0.835	0.014	61.3	<0.001

## Data Availability

Not applicable.

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
