# Peer review of "Age Assessment in Children and Adolescents by Measuring the Open Apices in Teeth: A New Sardinian Formula"

_dentistry, 2022, doi:10.3390/dj10040050_

Round 1
Reviewer 1 Report
Suggestions to Authors:
The study on the Cameriere method for the dental age estimation of young Sardinians is of interest, generally clearly explained and expressed, and appears worth publishing.
Some suggestions are however noted:
- In materials and methods, for the general reader there should be a clearer explanation about the variables used in the fitting expressions for predicted age. For example, the meaning of N0 is repeated multiple times but the variable s seems less clearly defined.
- The correlation quantities of R and R^2 are mentioned as recorded values but not reported in the results. Why not, considering that uncertainty is a part of the discussion?
- The conclusions suggest expanding the population sample to more than 300. Considering the current outcome from about 200 people, this seems rather low, and more convincing differences could require 600 or more? Can you justify the number of 300?
Reviewer 2 Report
After reviewing the manuscript “Age assessment in children and adolescents by measuring the open apices in teeth: a new Sardinian formula” according to the criteria for publication of the Dentistry Journal, we have made ​​the following assessment:
Review Criteria: After a brief bibliographical search, we have found that the lack of studies of estimation of the chronological age in Sardinian children and adolescents (a genetically isolated population), as well as the accuracy of the model proposed (Cameriere’s European and Italian formulae) in Sardinian children and adolescents makes this paper of a great interest.
Moreover, correct age assessment methods are especially important in the process of international adoption and in the case of immigrants without valid documents confirming their identity. In these cases, having specific formulas for particular populations is a matter of great importance.
Regarding de manuscript, the clarity in drafting, and its methodological approach is adequate, as well as the exposition of the information and results. Also, the bibliography is representative of the ideas of the paper.
With regard to the abstract or keywords, are also adequate and representative of the general idea of the study.
Due to the quality and clarity, our proposal is to publish without modifications.
Reviewer 3 Report
Line 24: Appears “variables s”, but there is not explained before what “s” means.
Line 23: “correlation with age except for NO and g”, but there is not explained before what “g” means.
Line 63: You affirm that third molars are not Good indicators, but it is not clear if it is because of a physiological, anthropological, or genetic regulation cause? It would be useful for a cite to clarify this question.
Line 112: “…were taken from patients attending an orthodontic treatment or for diagnostic and therapeutic purposes…”
From a forensic point of view, to identify an individual among the full population. Is it possible to use this data to be extrapolated to the general population? Are there no perceptible differences between the healthy population and the pathological population such as the one selected which could affect the teeth development and in consequence to the results interpretation of the study?
Line 123: “The study also was performed in accordance with the ethical standards laid down by the Declaration of Helsinki (Finland) developed by the World Medical Association (WMA) as a statement of ethical principles for medical research involving human subjects”.
Has the study passed through some kind of ethics committee? Is there a letter or certificate issued by said committee?
Line 270: “A possible limitation of this study to be counted is the small sample size (<300 subjects) of children present in our study population, which might have affected the nonsignificant difference in developmental stages of each left mandibular tooth between European, Italian and Sardinian children, and adolescents”.
Keep in mind that the population of Sardinia is just over 1.5 million people, compared to Italy, which is 60 million, I think that would be a way of justifying it.
Reviewer 4 Report
First of all, I would like to congratulate the authors for such interesting research, which is a challenge nowadays. Age assessment requirements in the living are increasing tremendously in the last few years, and the performed and validation of dental methods make this purpose easier to be solved. Furthermore, one of the most important challenges to deal with is the enormous variety of populations without studies of reference. In this research the authors worked on a sample of 202 OPT from the Sardinian population. By itself, it seems the sample es too small but the enormous interest of the Sardinian population which is isolated is enough to make it a current reference study for this population. In addition, the sample has a fantastic distribution and its very well represented in all the age groups.
The content information is well organized according to the different epigraphs and the format is thorough and very clear. The methodology is widely explained so the method can be easily reproducible. In addition, figures and tables are enough explanatory to understand the method.
However, some suggestions are exposed as follows and should be in consideration before be published:
- Abstract: Could authors include a small part of the state of art at the beginning of the abstract?
- M&M (measurements): Is it possible to add a brief explanation about Cameriere’s methodology to asses dental age? Some of the researchers are familiarized with these methods but perhaps other don’t. I miss a brief paragraph to detail the methods used.
Apart from that, I would like to encourage the authors to keep working on this line of research to improve the methodology of age estimation in the living.
Author Response
Please the Attachment

Reviewer 5 Report
Dear authors,
Congratulation for your new research in the field of Camerieres formula for dental age determination in Sardinian population.
